# Global depression in breast cancer patients: Systematic review and meta-analysis

**Akbar Javan Biparva[1], Samira Raoofi[2], Sima Rafiei[3], Maryam Masoumi[4], Maryam Doustmehraban[5], Farzaneh Bagheribayati[6], Elahe sadat Vaziri Shahrebabak[7], Zahra Noorani Mejareh[8], Saghar Khani[8], Bahare Abdollahi[8], Zahra Abedi Bafrajard[7], Shakila Sotoude Beidokhti[7], Seyedeh Fahimeh Shojaei[9], Mahdieh Rastegar[7], Fatemeh Pashazadeh Kan[7], Elmira Nosrati Sanjabad[7], Ahmad Ghashghaee[10]\***

**1** Student Research Committee, School of Health Management and Medical Informatics, Iranian Center of Excellence in Health Management, Tabriz University of Medical Sciences, Tabriz, Iran, **2** Department of Health Services Management, School of Health Management and Information Sciences, Iran University of Medical Sciences, Tehran, Iran, **3** Social Determinants of Health Research Center, Research Institute for Prevention of Non-Communicable Diseases, Qazvin University of Medical Sciences, Qazvin, Iran, **4** Clinical Research and Development Center, Qom University of Medical Sciences, Qom, Iran, **5** Social Determinants of Health Research Center, Qazvin University of Medical Sciences, Qazvin, Iran, **6** Cellular and Molecular Biology, University of Zanjan, Zanjan, Iran, **7** Student Research Committee, School of Nursing and Midwifery, Iran University of Medical Sciences, Tehran, Iran, **8** Student Research Committee, School of Medicine, Iran University of Medical Sciences, Tehran, Iran, **9** Firoozgar Clinical Research and Development Center (FCRDC), Iran University of Medical Sciences, Tehran, Iran, **10** The School of Medicine, Dentistry & Nursing, University of Glasgow, Glasgow, United Kingdom

\* Ahmad.ghashghaee1996@gmail.com

**Data Availability Statement:** All relevant data are within the paper and its Supporting Information files.

## Abstract

### Background

Breast cancer is known as one of the most common diseases among women, the psychological consequences of which are common in women and affect various aspects of their lives, so this study aims to investigate the prevalence of depression among women with breast cancer globally.

### Method

The present meta-analysis was performed by searching for keywords related to breast cancer and depression in 4 main databases: PubMed, Embase, Web of Sciences and Scopus in the period of January 2000 to November 2021 and the results of the study using R and CMA software were analyzed.

### Results

A total of 71 studies were selected in English and the results of the analysis showed that the prevalence of depression in women with breast cancer is 30.2%, with Pakistan having the highest (83%) prevalence of depression and Taiwan having the lowest (8.3%). And in the WHO regions, EMRO region had the highest (49.7%) rate and SEARO region had the lowest (23%) prevalence of depression. Also, with increasing age, the prevalence of depression among women with breast cancer increases.

**Funding:** The author(s) received no specific funding for this work.

**Competing interests:** The authors have declared that no competing interests exist.

## Conclusion

Community and family support for women with breast cancer, holding psychology and psychotherapy courses, lifestyle modifications and training in this area can be effective in preventing the reduction of the prevalence of depression, and given the pivotal role of women in family affairs, this This can be in line with the work of health system policymakers.

## Introduction

In 2018, about 18.1 million people worldwide were diagnosed with cancer, of which 9.6 million died from the disease. These figures are expected to double by 2040, with the maximum increase in low-to-middle-income countries, which account for more than two-thirds of the world's cancers [1]. The most common types of cancers diagnosed in different countries are lung and female breast cancer accounting for 11.6% of all cases, followed by colorectal cancers (10.2%) [1]. This common type of cancer among women has been known as the fifth leading cause of cancer death in both sexes between 2005 and 2015 [2].

According to previous studies, breast cancer causes more lost disability-adjusted life years (DALYs) and it often occurs during the middle age life with increasing rates in older ages [3]. Incidence rates of the disease are different considerably among various continents, from 27 per 100,000 in Middle Africa and Eastern Asia to 92 per 100,000 in Northern America [4]. Worldwide statistics on breast cancer published in 2016 revealed that Asia accounts for 44% of world's cancer deaths with 39% of total new diagnosed cases [5]. Thus, screening priority should be given to this type of cancer among women in all countries due to the fact that early-stage cancers are easier to treat and they have higher chance of survival [6].

Despite the improvements in screening, diagnostics and treatment of breast cancer, patients are still not properly screened and do not receive adequate social support, especially in developing countries [7]. Evidence has shown that support from close family members has a positive effect on patients' physical health and mental well-being, as well as their ability to adapt their living conditions to the chronic illness and associated symptoms including pain, difficulty in falling asleep, distress and depression [8]. Several studies revealed that depression is an important concern for breast cancer patients which is considerably related to physical deficit, disease severity, ill-health condition, poor performance and reduced survival [9–13]. A systematic review conducted by Pilevarzadeh et al. reported that the global prevalence of depression among breast cancer patients is 32.2% [14]. In fact, based on the results of various studies, the prevalence rate has been reported to be between 9.3 to 56 percent [15–19]. This range varied from 1.5% to 50% among women with breast cancer [17, 20]. Furthermore, if breast cancer coincides with depression, patients will experience more severe pain, extreme fatigue, decreased life expectancy and diminished quality of life (QOL) [21, 22]. In a recent meta-analysis, results affirmed that cancer patients with depressive symptoms had a 25% higher mortality rate [23]. In other words, even after receiving effective physical therapy, the existing stress and depression may remain and eventually lead to a lower recovery rate and decreased level of QOL [24, 25]. Despite the importance of the issue, most of the oncology settings focus on treatment procedures which mainly are based on physical signs and symptoms of patients whereas their psychological distress and mental well-being are frequently ignored. Thus, in order to obtain evidence-based data on the overall prevalence of depression among breast cancer patients worldwide and the associated factors, this systematic review and meta-analysis was

conducted to cover the existing research gap and define future research priorities on the importance of sustainable evidence-based psychosocial care for cancer survivors.

## Methods

### Registration and reporting

The systematic review was submitted with PROSPERO 2021 ID 309783. The related methods are in line with the guidelines of the Preferred Reporting Items for Systematic Reviews and Meta-Analyses (PRISMA) 2020 [26].

### Data bases and search terms

A comprehensive review of data bases including Embase, Scopus, PubMed, and Web of Science was conducted for original articles published in English from January 2000 to November 2021.

A total of 944 articles were retrieved from the initial search in different electronic databases. After removing the duplicates, a total number of 461 articles remained for further review. To check the data relevancy two independent researchers reviewed the titles and abstracts independently leading to 117 articles. Then the full texts of articles were deeply reviewed to confirm that the defined eligibility criteria were met properly. Accordingly, studies which incorporated data on depression prevalence among breast cancer patients or its determinants were considered for further review. Conference abstracts were also searched and the references of included articles were examined to be included as additional references. Finally, applying inclusion/exclusion criteria resulted in 71 studies which included in this study (Fig 1).

### Inclusion and exclusion criteria

Studies with quantitative data on depression prevalence among breast cancer patients and its determinants were included to find a set of articles based on the research keywords. Different types of observational studies containing cross-sectional, case-study, case-series, prospective, and cohort were involved. Additionally articles published in English between January 2000 and November 2021 were considered in the review. Exclusion criteria were other types of studies including interventional studies, case-control, reviews, letter to the editor, books, reports, and commentaries published in languages other than English and performed on patients with cancers other than breast cancer. Furthermore, studies with inadequate data on research questions, and those focusing on diagnosis or therapeutic approaches, and medication therapies were not included in the review.

### Quality assessment

To assess the risk of bias of included studies, The Newcastle-Ottawa Scale (NOS) was used. This checklist evaluates the quality of studies along with dimensions including case definition, selection of controls, comparability of cases and controls, and exposure/outcome in three main sections of exposure/outcome ascertainment, selection of study groups, and their comparability. Each study was evaluated for risk of bias by two independent researchers; in case of any discrepancy the consensus was achieved through consulting with a third party. The lowest and highest NOS scores for each of the evaluated articles could be in a range between 0 and 10, so that an article with score below four was mentioned to have a low level of quality [27].

### Data extraction

A data extraction form was used to enter the data of included studies by two independent investigators. The form included requisite information including first author's name,

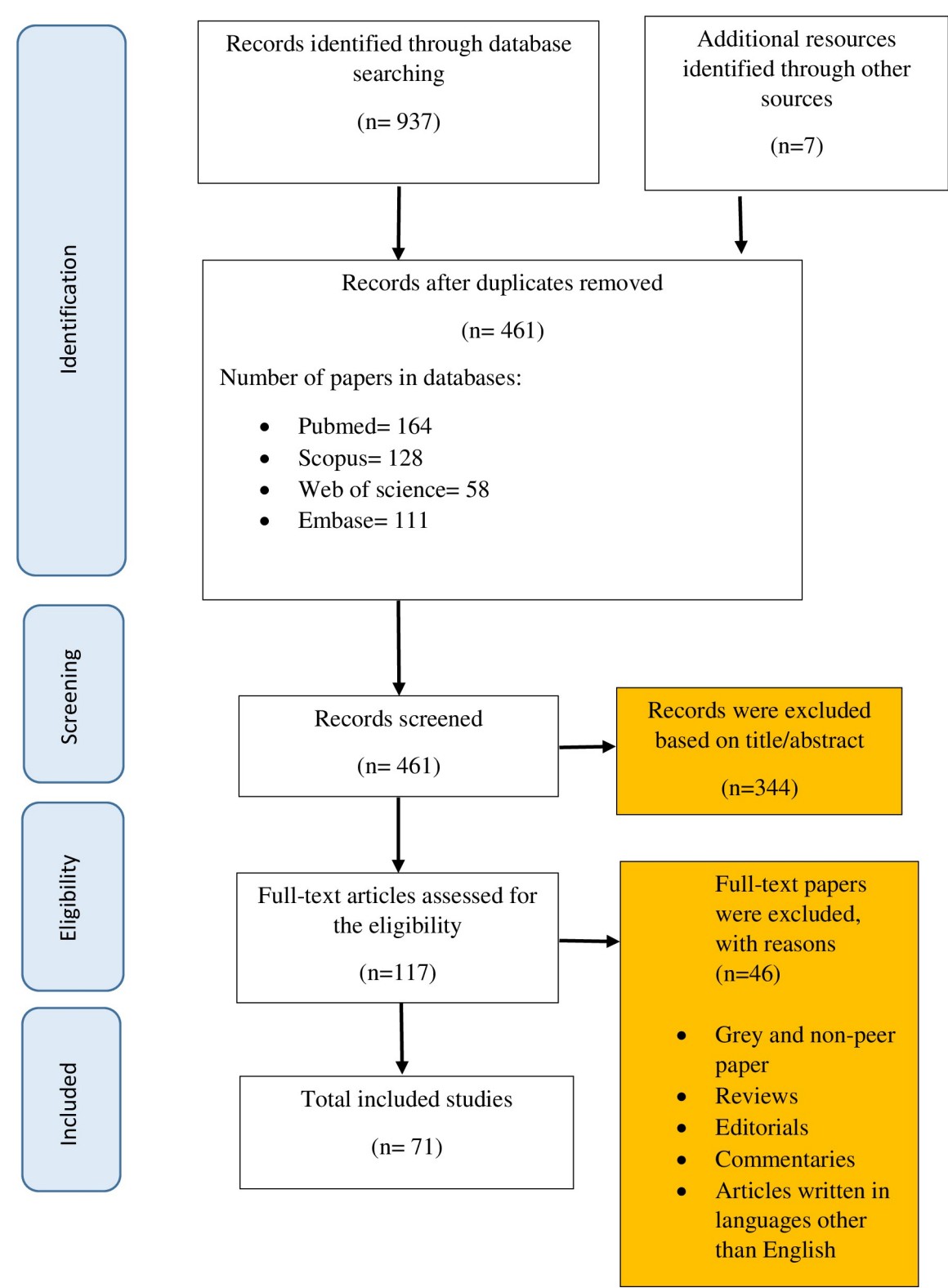

**Fig 1. Flow diagram of our review process (PRISMA).**

publication date and country, data gathering tool, study design, study population, sampling method, region based on WHO classification, risk of bias, outcome measures (prevalence of depression) and associated factors.

## Statistical analysis

The prevalence of depression was evaluated by random-effects model. Data were combined with the forest plot. The heterogeneity of preliminary studies was evaluated with $I^2$ test. In addition, subgroup analysis was used to determine heterogeneity based on different study settings and patients' socio-demographic characteristics. Meta-analysis was performed using Comprehensive Meta-Analysis and R software.

## Results

To report the findings of this review, Preferred Reporting Items for Systematic Reviews and Meta-analysis (PRISMA) guideline was employed [28]. After reviewing 71 articles published from January 2000 to November 2021, the total prevalence rate of depression among breast cancer patients was estimated at 30.2% (95% CI, 24–37.2) (Fig 2).

## Subgroup analysis for countries, continents and WHO regions

Regarding the results of meta-analysis, the highest rates of depression were respectively belonged to Pakistan at 83.9% (95% Cl, 73.6–89.5) and Greece at 50.03% (95% Cl, 38–62). On the other hand, the lowest rates of depression belonged to Taiwan at 8.3% (95% Cl, 5.7–12) and Canada 15.8% (95% Cl, 14.1–17.6) respectively (Table 1). In addition, related findings based on different continents showed that the highest depression prevalence was observed in Asia at 34.2% (95% CI, 23.2–47.2) and Africa at 32.4% (95% CI, 18.1–50.2); while the lowest prevalence rate was observed in South America at 17.3(95%CI, 13.2–22.3) (Table 1). The

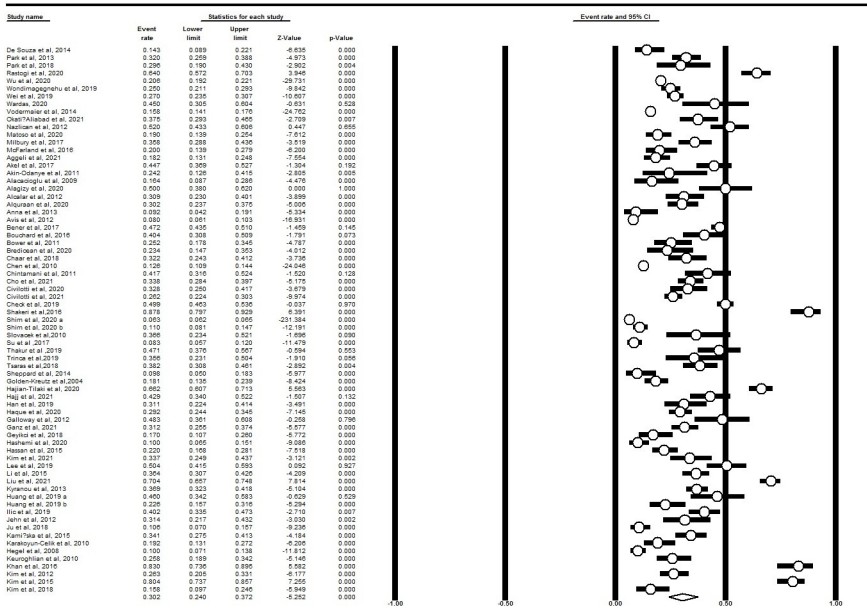

**Fig 2. The forest plot of depression in breast cancer patients.**

**Table 1. Meta-analysis based on countries, continents and WHO regions.**

| Groups | | Effect size and 95% interval | | | Test of null (2-Tail) | |
|---|---|---|---|---|---|---|
| | | Point estimate | Lower limit | Upper limit | Z-value | P-value |
| Countries | Brazil | 0.173 | 0.132 | 0.223 | -9.672 | 0.000 |
| | Canada | 0.158 | 0.141 | 0.176 | -24.762 | 0.000 |
| | China | 0.338 | 0.169 | 0.563 | -1.425 | 0.154 |
| | Egypt | 0.500 | 0.380 | 0.620 | 0.000 | 1.000 |
| | Ethiopia | 0.250 | 0.211 | 0.293 | -9.842 | 0.000 |
| | France | 0.322 | 0.243 | 0.412 | -3.736 | 0.000 |
| | Germany | 0.314 | 0.217 | 0.432 | -3.030 | 0.002 |
| | Greece | 0.272 | 0.121 | 0.503 | -1.938 | 0.053 |
| | India | 0.446 | 0.376 | 0.519 | -1.457 | 0.145 |
| | Iran | 0.494 | 0.191 | 0.801 | -0.031 | 0.975 |
| | Italy | 0.189 | 0.048 | 0.519 | -1.861 | 0.063 |
| | Jordan | 0.302 | 0.237 | 0.375 | -5.006 | 0.000 |
| | Lebanon | 0.429 | 0.340 | 0.522 | -1.507 | 0.132 |
| | Malaysia | 0.220 | 0.168 | 0.281 | -7.518 | 0.000 |
| | Multiple | 0.336 | 0.210 | 0.490 | -2.088 | 0.037 |
| | Nigeria | 0.242 | 0.126 | 0.415 | -2.805 | 0.005 |
| | Pakistan | 0.830 | 0.736 | 0.895 | 5.582 | 0.000 |
| | Poland | 0.374 | 0.282 | 0.477 | -2.397 | 0.017 |
| | Portugal | 0.356 | 0.231 | 0.504 | -1.910 | 0.056 |
| | Qatar | 0.472 | 0.435 | 0.510 | -1.459 | 0.145 |
| | Romania | 0.234 | 0.147 | 0.353 | -4.012 | 0.000 |
| | Serbia | 0.402 | 0.335 | 0.473 | -2.710 | 0.007 |
| | South Korea | 0.267 | 0.130 | 0.470 | -2.229 | 0.026 |
| | Taiwan | 0.083 | 0.057 | 0.120 | -11.479 | 0.000 |
| | Turkey | 0.275 | 0.172 | 0.409 | -3.155 | 0.002 |
| | United states | 0.258 | 0.189 | 0.343 | -5.129 | 0.000 |
| Continents | Africa | 0.324 | 0.181 | 0.509 | -1.868 | 0.062 |
| | Asia | 0.342 | 0.232 | 0.472 | -2.361 | 0.018 |
| | Europe | 0.289 | 0.240 | 0.344 | -6.954 | 0.000 |
| | North America | 0.249 | 0.186 | 0.325 | -5.793 | 0.000 |
| | South America | 0.173 | 0.132 | 0.223 | -9.672 | 0.000 |
| | Multiple | 0.313 | 0.172 | 0.499 | -1.966 | 0.049 |
| WHO Regions | AFRO | 0.249 | 0.212 | 0.291 | -10.233 | 0.000 |
| | AMRO | 0.238 | 0.181 | 0.306 | -6.620 | 0.000 |
| | EMRO | 0.497 | 0.372 | 0.623 | -0.045 | 0.964 |
| | EURO | 0.296 | 0.246 | 0.351 | -6.761 | 0.000 |
| | SEARO | 0.230 | 0.083 | 0.498 | -1.976 | 0.048 |
| | WPRO | 0.305 | 0.208 | 0.423 | -3.138 | 0.002 |
| | Multiple | 0.313 | 0.172 | 0.499 | -1.966 | 0.049 |

meta-regression results based on WHO regions demonstrated that the highest and lowest depression prevalence was observed in EMRO 49.7% (95% Cl, 37.2–62.3) and SEARO 23% (95% Cl, 8.3–49.8) (Table 1).

## Meta-analysis for different stages of treatment

To systematically enrich the review, we divided study participants into two groups of patients; those who were under the treatment procedure and those who has completed the treatment duration. According to the analysis, people in the latter group had lower rate of depression 25.7(95% CI, 16.9–36.9) compared with patients who were under the treatment procedure 32.6(95% CI, 27.6–38.1) (Table 2).

## Meta-regression based on publication year

The results of meta-regression analysis by the year of publication revealed that the depression prevalence which has been reported in the articles published in each year was on average 0.14% lower than the corresponding figure reported in previous years. Thus, as Fig 3 indicates there is a significant inverse relationship between depression prevalence and the passing of time (Fig 3).

## Sub-group analysis for age

A meta-regression for age showed that a unit of increase in patient's age increased the prevalence of depression by 0.57%. In fact a significant direct relationship between depression in breast cancer patients and their age was affirmed in this review (P-value<0.05) (Fig 3).

## Meta-regression by type of questionnaires

The instruments used in almost half of the studies were Hospital Anxiety and Depression (HADS), and Beck Depression Inventory (BDI); through which the depression prevalence was close to the overall reported prevalence rate at 31.2(95% CI, 25.4–37.7) and 32.8(95% CI, 23.2–44.1) respectively. On the other hand, 13 studies used researcher-made and mixed questionnaires (Table 3).

## Meta-regression for quality assessment

Results based on the quality assessment showed that more than half of the included studies (n = 57) had high quality, while 14 studies indicated medium-level of quality and no studies were of low quality (Table 3).

## Discussion

Feelings of depression, distress, and anxiety are among the reactions mostly evident among breast cancer patients throughout the diagnostic process and their disease progression. These patients feel exhausted and hopeless during the treatment process and play less active roles in social environments and gradually reduce their relationships with people in the community [7]. In this systematic review, the depression prevalence among breast cancer patients was found to be 30.2%, which could have negative effect on patients' adherence to treatment, and consequently diminish their QOL and overall survival. Nearly a similar prevalence of depression was reported from studies conducted in Asian countries [15, 29–31]. However, higher

**Table 2. Meta-analysis based on stage of treatment.**

| Groups | Number Studies | Effect size and 95% interval | | | Test of null (2-Tail) | |
|---|---|---|---|---|---|---|
| | | Point estimate | Lower limit | Upper limit | Z-value | P-value |
| Treated | 23 | 0.257 | 0.169 | 0.369 | -3.966 | 0.000 |
| Under treatment | 48 | 0.326 | 0.276 | 0.381 | -5.960 | 0.000 |

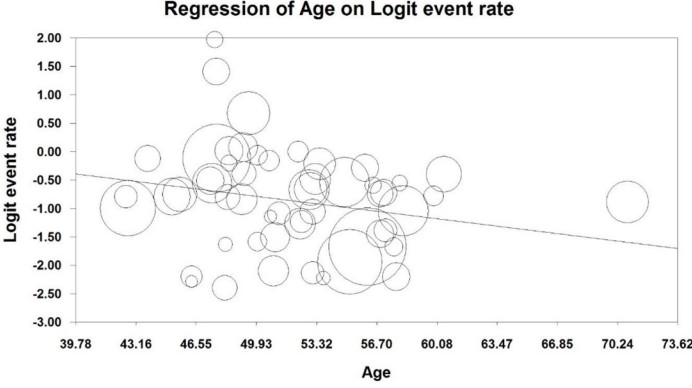
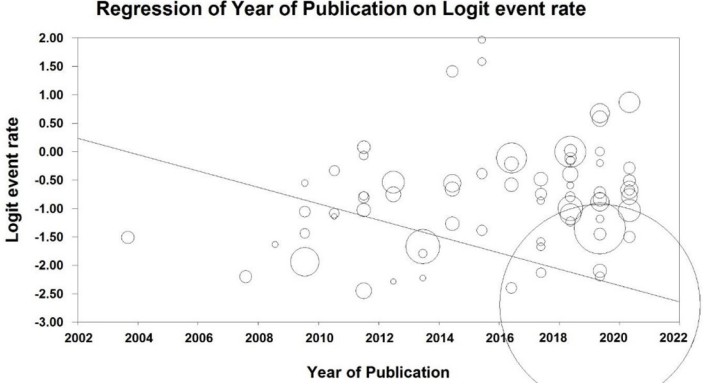

**Fig 3. Meta-regression based on year of publication and age.**

rate of depression was reported from Turkey (46%), South Africa, Mexico (43%) and Nigeria 40.3% which respectively used HADs, Center for Epidemiologic Studies Depression Scale (CESD) and Mini International Neuropsychiatric Interview (MINI) to assess depression [32–34]. One of the main justifications for this disparity might be due to the inclusion of newly diagnosed breast cancer patients in mentioned studies. In fact, patients who are newly diagnosed with cancer disease are more expected to have significantly higher rates of depressive symptoms. This finding has also been supported by a study conducted in China which discovered that the incidence rate of depression is much higher among newly diagnosed breast cancer patients [35]. Similarly according to our analysis, patients who were under the treatment procedure had higher rate of depression compared to those who has completed the treatment duration. Another reason for the differences in the prevalence of depression reported in various studies is related to the utilization of different tools in measuring patients' depression [36]. Dissimilar sample size, variation on the diagnostic tool, the geographic differences and the economic and socio-cultural differences between populations are among other contributing factors to this disparity [7].

Existing literature shows that despite a significant number of researches have been done in the area of depression among breast cancer patients; there are no clear data about the depression prevalence in different subgroups of patients. To provide necessary information about the issue, we conducted subgroup analysis to verify the prevalence of depression based on different determinants including age, stage of cancer, depression assessment tool, countries, continents,

**Table 3. Meta-analysis based on tools and quality of study.**

| Groups | | Number Studies | Effect size and 95% interval | | | Test of null (2-Tail) | |
|---|---|---|---|---|---|---|---|
| | | | Point estimate | Lower limit | Upper limit | Z-value | P-value |
| **Tools** | BDI | 18 | 0.328 | 0.232 | 0.441 | -2.934 | 0.003 |
| | CES-D | 9 | 0.275 | 0.186 | 0.386 | -3.748 | 0.000 |
| | DASS_21 | 2 | 0.154 | 0.063 | 0.329 | -3.365 | 0.001 |
| | HADS | 19 | 0.312 | 0.254 | 0.377 | -5.407 | 0.000 |
| | HAM-D | 3 | 0.305 | 0.122 | 0.457 | -2.335 | 0.020 |
| | PHQ-9 | 7 | 0.260 | 0.168 | 0.378 | -3.728 | 0.000 |
| | Other | 13 | 0.342 | 0.183 | 0.548 | -1.514 | 0.130 |
| **Quality of Study** | High | 57 | 0.329 | 0.232 | 0.377 | -4.764 | 0.000 |
| | Medium | 14 | 0.277 | 0.182 | 0.398 | -3.446 | 0.001 |

and WHO regions. In line with several studies, we found that the stage of breast cancer is significantly related to depression; so that patients in advanced stage of the disease revealed higher risk of depression compared to those in early stage cancer [37–39]. On the other hand, the higher incidence of depression immediately after the disease diagnosis and throughout the treatment process might be due to the fear of cancer recurrence and death anxiety, as well as adverse effects associated with medication and cognitive impairments associated with breast cancer therapies [18]. According to the importance of subject, the National Comprehensive Cancer Network (NCCN) has developed a guideline through which medical centers have been obligated to use appropriate screening tools to assess the mental status, level of stress or depression in cancer patients in order to facilitate further appropriate interventions for eliminating their emotional distress [40]. In a sample of 99 women who were under a follow-up study conducted by Vahdaninia et al., exhaustion, tenderness and pain were found to be considerably associated with depression within a year after treatment [41]. Similar studies also emphasized on strong associations between emotional distress in breast cancer patients and the self-reported pain or fatigue among them [42–44].

Concerning the different risk factors for depressive symptoms among breast cancer patients, after controlling other confounding factors, age was revealed to be considerably related to depression. In our review, the risk of depression augmented as the age increased, and women who were older than 30-years old were more likely to encounter depression compared to younger individuals. Similarly, a study conducted in the United States of America found that increasing age could enhance the risk of depression in women with breast cancer [45]. Another study in Lithuania also estimated that the risk of suffering from depression in breast cancer patients older than 55 years old would be increased by 2.25 times compared to their younger counterparts [46]. These findings are in contrary to studies conducted in some of the countries which reported that the incidence of mental disorders including depression is higher among younger patients [36, 47]. They added that younger women were at greater risk for psychological disorders than older ones. These young women often decline mastectomy and feel more fear of death which will bring them greater emotional distress and depression. They also may have special concerns about caring their children during the disease therapy and associated disruptions of family life [32, 47]. Possible explanation for such a considerable variation between our study results and existing literature might be the inclusion of middle-age women in our review and the fact that most cases of breast cancer occur in women older than 50 years of age who are at higher risk of depression due to their demographic characteristics [48].

In a systematic review and meta-analysis conducted to investigate the global prevalence of depression among breast cancer patients, the depression prevalence was found to be highest in the Eastern Mediterranean region and two times higher in middle-income countries compared to developed countries [14]. The same as existing literature, we found that depression among cancer patients is more prevalent in developing or underdeveloped countries. For instance, in a study conducted in Vietnam, the depression prevalence was 58%, and in Nepal the rate was estimated at 65% [49, 50]. Conversely, the prevalence reported in developed countries such as United Kingdom, Denmark, and Japan was considerably lower at 10% [51–54]. This disparity can be explained by dissimilar levels of quality services provided in healthcare institutions of different countries. Thus, the effectiveness of healthcare services provided in latter group of countries might clarify the reason for low prevalence of depression among cancer patients in these countries. This distinction can also reflect the differences between socio-economic factors between different populations which obviously highlight the importance of rendering supportive psychiatric interventions particularly in low or middle-income countries.

Meta-analysis based on year of publication revealed that the depression prevalence among cancer patients has a declining trend, which is in contrary to studies which have been

conducted with similar objectives. The decreasing prevalence in our review might be due to the dominant use of depression diagnostic instruments including HADS and BDI which to some extent underestimate the rate of depression than other tools due to their self-reporting nature. Similar studies affirmed this finding and considered higher depression prevalence in studies which applied diagnostic clinical interviews based on strict clinical criteria suitable for identifying depressive disorders [55].

## Limitations

There was a high level of heterogeneity in obtained results due to the various types of articles, different sampling methods and evaluation tools which have been used to assess depression among breast cancer patients. Therefore, these findings should be used with necessary caution to justify these limitations. Second, due to lack of data in included studies we could not consider type of treatment, and the relative side effects in sub-group analyses. Third, grey literature and unpublished manuscripts were not included in our review.

## Conclusion

The study findings affirm the higher prevalence of depression among breast cancer patients in developing countries compared to developed geographical areas. This should acknowledge the health policymakers, especially in underdeveloped countries about the necessity for providing higher quality healthcare services which considers anti-depression programs for cancer patients in an integrated healthcare system. In fact, psychological interventions can be regarded as effective strategies to improve both physical and psychological well-being of breast cancer patients. However, prior to psychiatric therapy, it is recommended to assess the depression prevalence of patients and recognize the determining factors to mental disorders. In this way, psychological supportive interventions could be suggested which can most likely meet the mental and emotional needs of patients.

## Supporting information

**S1 Checklist. PRISMA 2020 checklist.**
(DOCX)

**S1 Data.**
(XLSX)

## Author Contributions

**Conceptualization:** Zahra Noorani Mejareh, Mahdieh Rastegar, Ahmad Ghashghaee.

**Data curation:** Farzaneh Bagheribayati, Zahra Noorani Mejareh, Mahdieh Rastegar, Ahmad Ghashghaee.

**Formal analysis:** Farzaneh Bagheribayati, Zahra Noorani Mejareh, Mahdieh Rastegar, Ahmad Ghashghaee.

**Funding acquisition:** Mahdieh Rastegar.

**Investigation:** Farzaneh Bagheribayati, Saghar Khani, Bahare Abdollahi, Zahra Abedi Bafrajard, Seyedeh Fahimeh Shojaei, Ahmad Ghashghaee.

**Methodology:** Farzaneh Bagheribayati, Elahe sadat Vaziri Shahrebabak, Saghar Khani, Bahare Abdollahi, Zahra Abedi Bafrajard, Seyedeh Fahimeh Shojaei, Ahmad Ghashghaee.

**Project administration:** Saghar Khani.

**Resources:** Samira Raoofi, Elahe sadat Vaziri Shahrebabak, Zahra Abedi Bafrajard, Seyedeh Fahimeh Shojaei, Fatemeh Pashazadeh Kan, Ahmad Ghashghaee.

**Software:** Samira Raoofi, Elahe sadat Vaziri Shahrebabak, Shakila Sotoude Beidokhti, Seyedeh Fahimeh Shojaei, Ahmad Ghashghaee.

**Supervision:** Samira Raoofi, Sima Rafiei, Elahe sadat Vaziri Shahrebabak, Shakila Sotoude Beidokhti, Ahmad Ghashghaee.

**Validation:** Sima Rafiei, Shakila Sotoude Beidokhti, Fatemeh Pashazadeh Kan, Ahmad Ghashghaee.

**Visualization:** Sima Rafiei, Shakila Sotoude Beidokhti, Fatemeh Pashazadeh Kan, Ahmad Ghashghaee.

**Writing – original draft:** Akbar Javan Biparva, Sima Rafiei, Ahmad Ghashghaee.

**Writing – review & editing:** Akbar Javan Biparva, Sima Rafiei, Maryam Masoumi, Maryam Doustmehraban, Elmira Nosrati Sanjabad, Ahmad Ghashghaee.

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
