## [Decision Letter · Decision Letter 0]

27 Jan 2023

PONE-D-22-23966Global depression in Breast Cancer Patients: systematic review and meta-analysisPLOS ONE

Dear Dr. Ghashghaee,

Thank you for submitting your manuscript to PLOS ONE. After careful consideration, we feel that it has merit but does not fully meet PLOS ONE’s publication criteria as it currently stands. Therefore, we invite you to submit a revised version of the manuscript that addresses the points raised during the review process.

ACADEMIC EDITOR: The search period has been outdated by almost one year at the time of submission, please update the search and revise the analysis.  This is certainly an important topic, please address and provide the novelty and unique aspects in this study. Please response to the reviewers' comments and questions and revise the manuscript accordingly.  ==============================

We look forward to receiving your revised manuscript.

Kind regards,

Chien-Hsiang Weng, M.D., M.P.H.

Academic Editor

PLOS ONE

and https://journals.plos.org/plosone/s/file?id=ba62/PLOSOne_formatting_sample_title_authors_affiliations.pdf.

Reviewers' comments:

Reviewer's Responses to Questions

**Comments to the Author**

1. Is the manuscript technically sound, and do the data support the conclusions?

Reviewer #1: Yes

Reviewer #2: Partly

2. Has the statistical analysis been performed appropriately and rigorously? 

Reviewer #1: Yes

Reviewer #2: Yes

3. Have the authors made all data underlying the findings in their manuscript fully available?

Reviewer #1: Yes

Reviewer #2: No

4. Is the manuscript presented in an intelligible fashion and written in standard English?

Reviewer #1: Yes

Reviewer #2: Yes

5. Review Comments to the Author

Reviewer #1: Dear Editor

This is a good manuscript reviewing “Global depression in Breast Cancer Patients: systematic review and meta-analysis”. The subject of the manuscript is fully consistent with the aims and scope of the journal « PLOS ONE». The research methodology is fully consistent with the aims declared by the authors. Their conclusions are also consistent with the set goals, however, some issues need to be reconsidered:

- Please explain all abbreviations in the abstract and manuscript.

Abstracts

1) include data sources; study eligibility criteria, participants, and interventions; study evaluation and synthesis methods; results; limitations; conclusions and implications of key findings;

2) The search was performed more than 6 month ago so I think authors should update their search

3) Keywords: are these keywords are Mesh terms? Word that serves as a keyword, as to the meaning of that condition must be a Mesh term

Introduction

The Introduction needs adjustments in order to answer these questions:

- What are the uncertainties and conflicts that underlie the hypotheticals?

- How important is the evidence of studies for the healthy individuals and patients?

- What is the focused clinical question your review will address?

- There are some sentences that are difficult to understand, and the paper needs an English reviewer. Please edit.

Methods

-List and define all variables for which data were sought (e.g., PICOS, funding sources) and any assumptions and simplifications made.

- I suggest not including in the search strategy the outcome`s terms. The search should have more sensitivity than specificity.

-Why authors a priori used a random-effects model for their analysis? Where there any signs to expect significant heterogeneity from the studies included?

.

- it is not necessary to mention all keywords and syntaxes in the method section, authors can refer to main keywords in the methods and present full search strategy and syntaxes in a supplemental file.

- The search was performed more than 6 month ago so I think authors should update their search

- What are these key words? It is appears that they are suitable for just MEDLINE search, so what about other databases? I cannot search SCOPUS with these syntaxes. If the authors use different syntaxes, they should present them, if they don’t at least present keyword in plain form not in a specific database format.

Discussion

Authors should also acknowledge some serious limitations of the study:

1. significant heterogeneity was encountered perhaps due to various regimens, doses, duration, center settings, populations enrolled etc. calling for cautious interpretation of the results. This is a serious limitation and should be included because it may significantly undermine the validity of results.

2. many of the studies suffer from significant sources of bias and this should be also taken into consideration

3. the effect in many occasions was assessed by very few studies; thus, the evidence to support it is low and should be mentioned.

Reviewer #2: To Dear Author,

Cancer is well-known associated with depression. The association have been discussed in many articles previously. For example, "Pilevarzadeh, M., Amirshahi, M., Afsargharehbagh, R., Rafiemanesh, H., Hashemi, S. M., & Balouchi, A. (2019). Global prevalence of depression among breast cancer patients: a systematic review and meta-analysis. Breast cancer research and treatment, 176, 519-533." and the topic is almost the same with your manuscript.

I suggest that your manuscript should include some novel idea about association between breast cancer and depression. In addition, provider supplemental for more detail about your systematic review and meta-analysis.

Best regards,

6. PLOS authors have the option to publish the peer review history of their article (what does this mean?). If published, this will include your full peer review and any attached files.

Reviewer #1: No

Reviewer #2: No

---

## [Author Response · Author response to Decision Letter 0]

8 Mar 2023

Response to reviewers

Editor-in-Chief 

PLOS ONE

Art. No: PONE-D-22-23966

Please find the revised version of our manuscript “Global Depression in Breast Cancer Patients: systematic review and meta-analysis": which we would like to resubmit for publication as a review article in PLOS ONE.

Your comments and those of the reviewers were highly insightful and enabled us to greatly improve the quality of our manuscript. In the following pages there are our point-by-point responses to each of the comments of the reviewers as well as your own comments. Also declaration section revised and completed according to journal guideline. 

Revisions in the text are shown using green highlight for additions. We hope that the revisions in the manuscript and our accompanying responses will be sufficient to make our manuscript suitable for publication in PLOS ONE.

We shall look forward to hearing from you at your earliest convenience. 

Yours sincerely, 

*Corresponding Author: Ahmad Ghashghaee

Email: ahmad.ghashghaee1996@gmail.com

Reviewer Number Original comments of the reviewer Reply by the author(s)

Reviews 1 Abstracts

1) include data sources; study eligibility criteria, participants, and interventions; study evaluation and synthesis methods; results; limitations; conclusions and implications of key findings;

2) The search was performed more than 6 month ago so I think authors should update their search

3) Keywords: are these keywords are Mesh terms? Word that serves as a keyword, as to the meaning of that condition must be a Mesh term 1) Since there is word limitation of Abstract, so I cannot add all the materials into this section. However some items has been added.

2) It is impossible to update the research because it is a global systematic review and it will take another 6 month. Also based on evidence, less than two years would be acceptable for a massive systematic reviews like this.

3) Keywords have been changed

Reviews 1 Introduction

The Introduction needs adjustments in order to answer these questions:

- What are the uncertainties and conflicts that underlie the hypotheticals?

- How important is the evidence of studies for the healthy individuals and patients?

- What is the focused clinical question your review will address?

- There are some sentences that are difficult to understand, and the paper needs an English reviewer. Please edit. This section has been revised.

Reviews 1 Methods

-List and define all variables for which data were sought (e.g., PICOS, funding sources) and any assumptions and simplifications made.

- I suggest not including in the search strategy the outcome`s terms. The search should have more sensitivity than specificity.

-Why authors a priori used a random-effects model for their analysis? Where there any signs to expect significant heterogeneity from the studies included?

.

- it is not necessary to mention all keywords and syntaxes in the method section, authors can refer to main keywords in the methods and present full search strategy and syntaxes in a supplemental file.

- The search was performed more than 6 month ago so I think authors should update their search

- What are these key words? It is appears that they are suitable for just MEDLINE search, so what about other databases? I cannot search SCOPUS with these syntaxes. If the authors use different syntaxes, they should present them, if they don’t at least present keyword in plain form not in a specific database format. 1) All the variables have been listed

2) Search strategy has been deleted

3) Yes, Since it is a global systematic review and we included various studies for analyzing, random-effect would be the best method for avoiding heterogeneity.

Reviews 1 Discussion

Authors should also acknowledge some serious limitations of the study:

1. significant heterogeneity was encountered perhaps due to various regimens, doses, duration, center settings, populations enrolled etc. calling for cautious interpretation of the results. This is a serious limitation and should be included because it may significantly undermine the validity of results.

2. many of the studies suffer from significant sources of bias and this should be also taken into consideration

3. the effect in many occasions was assessed by very few studies; thus, the evidence to support it is low and should be mentioned. The Limitation section has been added.

---

## [Decision Letter · Decision Letter 1]

18 Apr 2023

PONE-D-22-23966R1Global depression in Breast Cancer Patients: systematic review and meta-analysisPLOS ONE

Dear Dr. Ghashghaee,

Thank you for submitting your manuscript to PLOS ONE. After careful consideration, we feel that it has merit but does not fully meet PLOS ONE’s publication criteria as it currently stands. Therefore, we invite you to submit a revised version of the manuscript that addresses the points raised during the review process.

ACADEMIC EDITOR:Please address and response to the comments from Review #2.Please ensure that your decision is justified on PLOS ONE’s publication criteria and not, for example, on novelty or perceived impact.

We look forward to receiving your revised manuscript.

Kind regards,

Chien-Hsiang Weng, M.D., M.P.H.

Academic Editor

PLOS ONE

Journal Requirements:

Reviewers' comments:

Reviewer's Responses to Questions

**Comments to the Author**

1. If the authors have adequately addressed your comments raised in a previous round of review and you feel that this manuscript is now acceptable for publication, you may indicate that here to bypass the “Comments to the Author” section, enter your conflict of interest statement in the “Confidential to Editor” section, and submit your "Accept" recommendation.

Reviewer #1: (No Response)

Reviewer #2: All comments have been addressed

2. Is the manuscript technically sound, and do the data support the conclusions?

Reviewer #1: (No Response)

Reviewer #2: No

3. Has the statistical analysis been performed appropriately and rigorously? 

Reviewer #1: (No Response)

Reviewer #2: Yes

4. Have the authors made all data underlying the findings in their manuscript fully available?

Reviewer #1: (No Response)

Reviewer #2: Yes

5. Is the manuscript presented in an intelligible fashion and written in standard English?

Reviewer #1: (No Response)

Reviewer #2: Yes

6. Review Comments to the Author

Reviewer #1: The revision by authors and correction that they made of the manuscript was satisfactory and I have no more concerns

Reviewer #2: To Dear Author,

Cancer is well-known associated with depression. The association have been discussed in many articles previously. For example, "Pilevarzadeh, M., Amirshahi, M., Afsargharehbagh, R., Rafiemanesh, H., Hashemi, S. M., & Balouchi, A. (2019). Global prevalence of depression among breast cancer patients: a systematic review and meta-analysis. Breast cancer research and treatment, 176, 519-533." and the topic is almost the same with your manuscript.

I suggest that your manuscript should include some novel idea about association between breast cancer and depression. In addition, provider supplemental for more detail about your systematic review and meta-analysis.

In addition, for different kinds of depression such as major depressive disorder, dysthymic disorder, and other depressive disorder, combined medications and psychotherapy are the standard treatment and should be mentioned in the introduction and conclusion section.

Best regards,

7. PLOS authors have the option to publish the peer review history of their article (what does this mean?). If published, this will include your full peer review and any attached files.

Reviewer #1: No

Reviewer #2: No

---

## [Author Response · Author response to Decision Letter 1]

27 Apr 2023

Firstly, our search period is from 2000-2021 which means we had collected to more years in comparison with mentioned article. Our exclusion and inclusion criteria are different. Also we have result about depression in bearst cancer patients based on Age, stage of treatment and countries and continets which is our novelty.

This issue was not among the main objectives of our study. Our main focus in this article was the prevalence rate in different territories and the relationship between depression and the main variables.

---

## [Editor Report · Decision Letter 2]

17 May 2023

PONE-D-22-23966R2Global depression in Breast Cancer Patients: systematic review and meta-analysisPLOS ONE

Dear Dr. Ghashghaee,

Thank you for submitting your manuscript to PLOS ONE. After careful consideration, we feel that it has merit but does not fully meet PLOS ONE’s publication criteria as it currently stands. Therefore, we invite you to submit a revised version of the manuscript that addresses the points raised during the review process.

We look forward to receiving your revised manuscript.

Kind regards,

Chien-Hsiang Weng, M.D., M.P.H.

Academic Editor

PLOS ONE

Journal Requirements:

Additional Editor Comments:

1) The authors have addressed the comments from the reviewers and revised accordingly.

2) Please have English as native language to edit the entire manuscript.

3) Authors should make sure all the fonts and sizes are the same throughout the manuscript.

---

## [Author Response · Author response to Decision Letter 2]

1 Jun 2023

All requirements have been addressed.

Also a native speaker has revised the language (Author name: Elmira Nosrati Sanjabad)

---

## [Editor Report · Decision Letter 3]

5 Jun 2023

Global depression in Breast Cancer Patients: systematic review and meta-analysis

PONE-D-22-23966R3

Dear Dr. Ghashghaee,

We’re pleased to inform you that your manuscript has been judged scientifically suitable for publication and will be formally accepted for publication once it meets all outstanding technical requirements.

Kind regards,

Chien-Hsiang Weng, M.D., M.P.H.

Academic Editor

PLOS ONE
---

## [Editor Report · Acceptance letter]

17 Jul 2023

PONE-D-22-23966R3 

Global Depression in Breast Cancer Patients: systematic review and meta-analysis 

Dear Dr. Ghashghaee:

I'm pleased to inform you that your manuscript has been deemed suitable for publication in PLOS ONE. Congratulations! Your manuscript is now with our production department. 

Kind regards, 

on behalf of

Professor Chien-Hsiang Weng 

Academic Editor

PLOS ONE